# Eosinopenia as a predictor of clinical outcomes in hospitalized patients with community-acquired pneumonia: A retrospective cohort study

**Wigdan Farah**[1]*, **Zhen Wang**[2], **Ognjen Gajic**[1], **Yewande E. Odeyemi**[1]

**1** Division of Pulmonary and Critical Care Medicine, Mayo Clinic, Rochester, Minnesota, United States of America, **2** Evidence-Based Practice Center, Rochester, Mayo Clinic, Rochester, Minnesota, United States of America

* Farah.Wigdan@mayo.edu

## Abstract

Eosinopenia has been reported as a predictor of unfavorable outcomes and a marker of severity in bacterial infections. We describe the association between eosinopenia and clinical outcomes in hospitalized patients with CAP. We conducted a retrospective study of hospitalized adult patients with community-acquired pneumonia at a large US academic medical center from January 2009 to December 2019. We collected data on patient demographics, disease severity, comorbidities, smoking history, inflammatory markers, blood eosinophil levels, mortality, length of hospital stay, and need for intensive care unit (ICU) or mechanical ventilation. According to blood eosinophil count, patients were grouped as eosinopenic (<50/μL) and non-eosinopenic (≥50/μL) based on prior studies. Analysis was performed using nonparametric Wilcoxon rank-sum test for continuous variables and the chi-square test for categorical variables. A logistic regression analysis with robust standard errors was used to assess the associations between eosinopenia and patient centered outcomes (in-hospital mortality, 30-day mortality, length of hospital stay, need for mechanical ventilation support, vasopressor support and ICU admission). Of the 3285 patients with CAP infection included in our analysis, 1304 (39.70%) were eosinopenic. Age, gender, race, and smoking status were similar between the two groups. The eosinopenic group had significantly higher inflammatory markers as measured by C-reactive protein (CRP), and higher disease severity scores., After adjusting for disease severity, chronic obstructive pulmonary (COPD), and CRP there was no significant difference in hospital mortality (odds ratio [OR] 2.16, 95% confidence interval [CI] 0.68-6.8), ICU admission (OR: 1.21, 95% CI: 0.83-1.79), invasive and non-invasive ventilatory support (OR: 1.21, 95% CI: 0.52-2.81). Contrary to previously published data, our analysis did not demonstrate an association between eosinopenia and increased mortality risk in hospitalized patients with CAP highlighting the complexity of CAP prognosis.

**Data availability statement:** All relevant data are within the manuscript and its Supporting Information files.

**Funding:** this research was suported by grant number K23HL168212 (National Institute of Health, recip-ient: YO), Robert D. and Patricia E. Kern Center for the Science of Health Care Delivery, Mayo Clinic (recipient: YO) and Robert A. Winn Diversity in Clinical Trials Career Development Award (recipient: YO). The funders had no role in study design, data collection, analysis, decision to publish, or preparation of the manuscript.

**Competing interests:** The authors have declared that no competing interests exist.

## Introduction

Community-Acquired Pneumonia (CAP) is one of the leading cause of morbidity and mortality worldwide [1–3]. Characterized by acute inflammation of the lung parenchyma and a wide range of clinical presentations, CAP imposes a substantial burden on healthcare systems, often necessitating hospitalization, intensive medical intervention and increased risk of rehospitalization [4–7].While advances in antimicrobial therapy and diagnostic techniques have improved the management of CAP, a substantial proportion of patients experience adverse clinical outcomes [2].

Over the years, clinical judgment has been the primary tool utilized by physicians to predict the severity of diseases and direct care. However, various studies have shown that relying solely on clinical judgment may result in both overestimation and underestimation of the severity of community-acquired pneumonia (CAP), leading to inappropriate hospitalization of mild cases or inadequate interventions for high-risk patients [8]. Therefore, several prognostication scores specific to community-acquired pneumonia (CAP) have been developed, including the pneumonia severity index (PSI), CURB65 score, and the Infectious Disease Society of America (IDSA) and American Thoracic Society (ATS) criteria for severe CAP [8].

Although these tools have demonstrated a remarkable ability to predict mortality and identify low-risk patients who can be treated on an outpatient basis, their ability to identify high-risk patients who may experience clinical deterioration is suboptimal, Additionally, using these scores in day-to-day clinical practice can be cumbersome [8,9]. As a result, various laboratory parameters, such as C-reactive protein (CRP), procalcitonin, blood urea nitrogen (BUN), neutrophil-lymphocyte ratio, and pro-adrenomedullin, have been employed to evaluate disease severity and predict clinical outcomes in community-acquired Pneumonia (CAP) [8–14].

Over the recent years, eosinopenia, defined as a reduced eosinophil count (EC), has garnered attention as a potential prognostic biomarker in various infectious and inflammatory conditions including sepsis, acute respiratory distress syndrome (ARDS), acute exacerbations of chronic obstructive pulmonary disease, and other infectious diseases [15–20]. Several mechanisms have been proposed to explain the association between eosinopenia and adverse clinical outcomes, including dysregulated immune responses, increased disease severity, and corticosteroid treatment [15].

In contrast to the robust literature on the prognostic value of eosinopenia in other clinical conditions, evidence on the prognostic role of eosinopenia in hospitalized patients with community acquired pneumonia remains limited.

A study by Mao et al [17]. demonstrated an association between eosinopenia and increased risk of long-term mortality in hospitalized patients with CAP but with a concomitant diagnosis of acute exacerbation of COPD. Notably, there was a significant difference in the total steroid exposure between the eosinopenic and non-eosinopenic groups in this study. Similar results were also reported in a second study by Echevarria et al [21] which focused on a cohort of hospitalized patients with a diagnosis of CAP, excluding patients with an underlying diagnosis of chronic respiratory disease in an attempt to address the effect of steroids on eosinophils. Although both studies reported significant increased CRP in the eosinopenic group compared with the non-eosinopenic group, suggesting a higher severity of illness in the eosinopenic group, the CRP levels were not considered in the analysis. Moreover, both studies were limited by the sample size.

This retrospective analysis aims to describe the association between eosinopenia and in-hospital clinical outcomes in patients admitted with CAP taking into consideration steroid exposure and CRP measurement.

## Materials and methods

### Study design and subject

This study is a retrospective analysis examining the association between eosinopenia and clinical outcomes of adult patients (18 years or older) hospitalized with community-acquired pneumonia (CAP) from January 2009 to December 2019 at a large US academic medical center. Subjects were included if they had eosinophil levels measured within 24 hours of hospital admission and had not received any steroid treatment prior to the test. Patients who declined the use of their medical records for research purposes were excluded Fig 1. This study was determined to be exempt by the Institutional Review Board (IRB application #17-011140). The Strengthening the Reporting of Observational Studies in Epidemiology guidelines were followed in the design and reporting of this observational study [22].

In compliance with guidelines for reporting retrospective studies involving medical records or archived samples, the electronic medical record was accessed on 11/02/2021 for research purposes to identify patients who were admitted between 01/01/2009 to 12/31/2019 and create non identified dataset that was utilized afterward for analysis. The authors did not have access to any information that could identify individual participants during or after data collection.

### Patient selection and data collection

The electronic medical record was utilized to identify a cohort of adult patients who were hospitalized with community-acquired pneumonia [23]. This was identified on admission utilizing the International Classification of Diseases, 9th, and 10th Edition (ICD 9 and ICD 10), codes 481-486 and J18 coupled with searches of documentation of CAP in the diagnosis

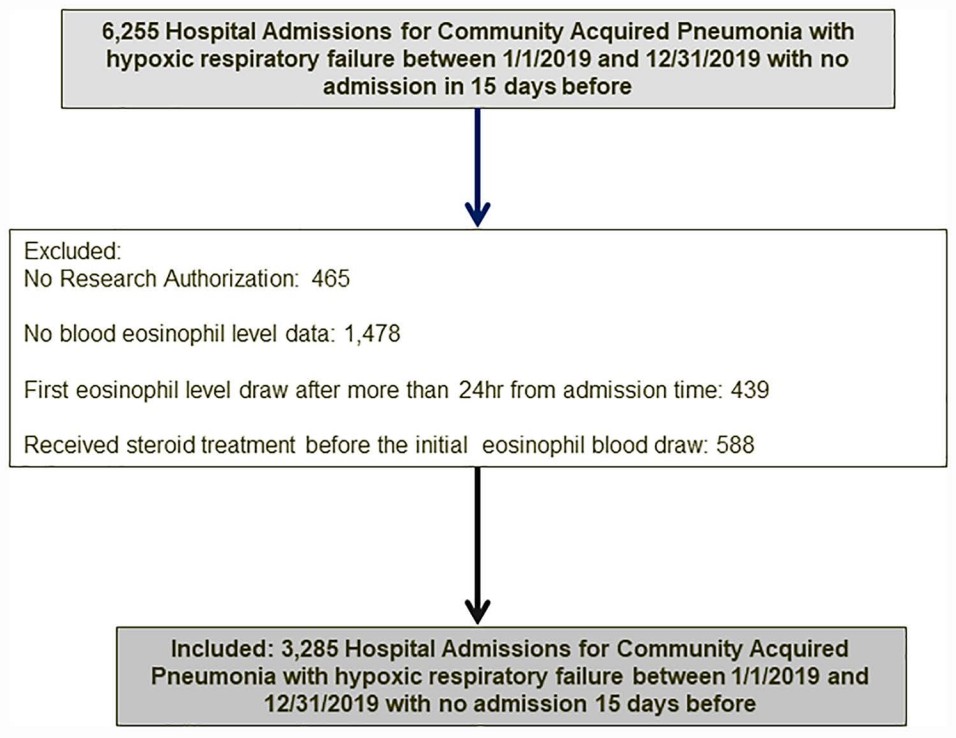

**Fig 1. Cohort flow chart illustrating the inclusion and exclusion of participants into the study.**

section of notes on the admission day. All patients were treated with guideline-recommended antibiotics based on institutional pneumonia order sets. However, information about specific antibiotics was not collected in this dataset.

The cohort dataset (S1 File) included demographic, clinical and laboratory data, including age, gender, race, smoking status, and other pertinent factors to characterize the study population. Various metrics and scoring systems used to assess disease severity, such as the CURB-65 score, Pneumonia Severity Index (PSI), Acute Physiology and Chronic Health Evaluation-III (APACHE-III) and sequential organ failure assessment (SOFA) scores at admission were employed to stratify patients based on the severity of their pneumonia. Various comorbid conditions including, but not limited to, chronic obstructive pulmonary disease (COPD), congestive heart failure, cardiovascular diseases, diabetes mellitus, and others were collected to evaluate their influence on outcomes.

Laboratory data on blood eosinophil counts were obtained within 24hours from admission, and patients were categorized into two groups based on prior studies [17,24]: eosinopenic (<50/μL) and non-eosinopenic (≥50/μL). Data on C-reactive protein was obtained within 48 hours.

## Outcomes

The primary outcome was in-hospital mortality. Secondary outcomes included hospital length of stay, intensive care unit admission, need for noninvasive ventilation and invasive mechanical ventilation support, need for vasopressor support, and 30-day mortality. In-hospital mortality was defined as all-cause mortality occurring during the hospital stay, while 30-day mortality encompassed all-cause mortality within 30 days from admission. Length of hospital stay was defined as the duration of each patient's hospitalization from admission to discharge.

## Statistical analysis

Data were summarized as median and IQR for continuous variables and, absolute and relative frequencies (%) for categorical variables. Shapiro-Wilk and Shapiro–Francia test of normality were performed to assess if the continuous variables followed a normal distribution. Comparisons between the eosinopenic and non-eosinopenia groups were performed using a nonparametric Wilcoxon rank-sum test for continuous variables that did not meet normal distribution and the chi-square test for categorical variables. A logistic regression analysis, employing robust standard errors, was performed to explore the associations between eosinopenia and the identified clinical outcomes. Robust standard errors were utilized to mitigate potential biases and confounding factors, ensuring the reliability and validity of the findings. The models were adjusted for the effects of pneumonia severity index and COPD: both chosen a priori. To evaluate the prognostic accuracy of various logistic regression models, cutoff values were determined through the analysis of the Receiver Operating Characteristic (ROC) curve, aiming to establish the optimal level of discrimination. The ROC analysis was employed to assess the diagnostic accuracy of the different parameters. The performance of these parameters was quantified using the Area Under the Receiver Operating Characteristic Curve (AUROC), where an AUROC of 1.0 signifies perfect discrimination, and 0.5 indicates chance-level discrimination.

Subgroup analysis was performed for the age group under 65 and for the group with available CRP level by adding continuous CRP level to the regression model. Additionally steroid use and early steroid use (defined as steroid use within 24 hours) were added to the regression models to evaluate the impact of eosinopenia and corticosteroid use on mortality outcomes. Sensitivity analyses were conducted using varying eosinophil level thresholds to determine

the impact of different eosinophil level on clinical outcomes, i.e., 50, 100, 150 and 200 mg/L in addition to the use of eosinophil as a continuous variable. Statistical analyses were performed using a standard software package (Stata Statistical Software: Release 16. College Station, TX: Stata Corp LLC.). Two-tailed P < 0.05 was considered as statistically significant.

## Results

Of the 3285 patients with CAP infection included in our analysis, 1304 (39.70%) were eosinopenic. Table 1 illustrates the distribution of baseline characteristics across both groups. Both groups had similar distribution in gender, race, and smoking status.

The eosinopenic group had a higher WBC, neutrophil count, blood glucose, BUN, procalcitonin, NT-proBNP, CRP, lactate, and sodium levels than the non-eosinopenic group (p < 0.05).

Patients with eosinopenia at admission were more likely to present with severe illness indicated by differences in physical exam findings (pulse rate, systolic blood pressure, diastolic blood pressure, respiratory rate, temperature and altered mental status) and higher disease severity scores such as PSI score, CURB65 score, and Apache III score (p < 0.05). Analysis of comorbidities demonstrated significant differences between the groups, with higher rate of congestive heart failure (CHF), asthma and overall Charlson comorbidity Index among the non-eosinopenic group Table 1.

### Outcomes

The analysis of outcomes revealed no statistically significant difference in in-hospital mortality (odds ratio [OR]: 1.39, 95% confidence interval [CI]: 0.90-2.2) and 30-day mortality (OR: 0.89, 95%CI: 0.66-1.19) with eosinopenic status. Also, no significant difference was observed in the need for non-invasive ventilation (NIV) ([OR]: 1.2, 95% CI: 0.94-1.54) and need for invasive mechanical ventilation (IMV) (OR: 1.3, 95%CI: 0.96-1.75). However, patients with eosinopenia exhibited a significantly increased likelihood of concurrently requiring both NIV and IMV (OR: 2.07, 95%CI: 1.45-2.95). Additionally, the presence of eosinopenia was significantly associated with increased need of ICU admission (OR: 1.55, 95%CI: 1.33-1.79), higher likelihood of requiring vasopressors (OR: 1.26, 95%CI: 1.6-1.49) and longer length of hospital stay (Mean Difference: 0.56, 95% CI 0.16-0.95) Table 2.

Multivariable analysis model adjusting for disease severity, and chronic obstructive pulmonary disease (COPD) again demonstrated an increased concurrent need for both NIV and IMV, ICU admission, and a lower risk for 30-day mortality among the eosinopenic group compared to non-eosinopenic group (Table 2). However, no significant difference in outcomes was found between the groups in the subgroup analysis for patients under 65 years old or when CRP level was added to the model in the subgroup of patients with available CRP (S1 and S2 Tables). Adjusting for steroid treatment showed an increased risk for 30-day mortality in the eosinopenic group compared with the non-eosinopenic group., On the other hand, a lower risk for 30-day mortality was observed after adjusting for early steroid treatment (within 24 hours) (S3 and S4 Tables). The sensitivity analysis showed similar results when using eosinophil as a continuous variable or different levels of eosinophil.

When comparing the prognostic accuracy of eosinopenia, pneumonia severity index, and the combined model of pneumonia severity index and eosinopenia, the combined model exhibited the highest discriminative value for 30-day mortality with an AUROC of 0.743 (95% CI = 0.71–0.77). Following closely were the pneumonia severity index (AUROC = 0.737, 95% CI = 0.71–0.77) and eosinopenia (AUROC = 0.514, 95% CI = 0.48–0.55) for 30-day mortality (p < 0.0001). Similarly, for a composite of secondary outcomes (including the need for

**Table 1. Demographic and clinical characteristics in hospitalized patients with community-acquired pneumonia by eosinopenia status.**

| Variables | Eosinopenic N: 1304(39.70%) | Non-eosinopenic N: 1981(60.3%) | P-value |
|---|---|---|---|
| Demographic Characteristics | | | |
| Age (Median;IQR) | 79 (22.5) | 78 (22) | 0.3 |
| Age >65 | 1019(78.1%) | 1523(76.9%) | 0.4 |
| Female | 586(44.9%) | 911(46%) | 0.6 |
| Smoking | 653(58.4%) | 958(59%) | 0.8 |
| Race | | | 0.5 |
| White | 1215(93.2%) | 1860(93.9%) | |
| Black | 18(1.4%) | 27(1.4%) | |
| Hispanic or Latino | 21(1.6%) | 32(1.6%) | |
| Asian | 18(1.4%) | 18(0.9%) | |
| American Indian | 7(0.5%) | 4(0.2%) | |
| Other | 25(1.9%) | 40(2.0%) | |
| Comorbidity | | | |
| Charlson Score (Median; IQR) | 7(5) | 7(5) | <0.01 |
| Asthma | 161(12.4%) | 312(15.8%) | <0.01 |
| CHF | 367(28.1%) | 642(32.4%) | <0.01 |
| COPD | 398(30.5%) | 632(31.9%) | 0.4 |
| Liver Disease | 14(1.1%) | 29(1.5%) | 0.4 |
| Laboratory Results | | | |
| BUN | 20(17) | 19(14) | <0.01 |
| CRP | 102.8(147.7) | 69.8(47.6) | <0.0001 |
| D-dimer | 45(410) | 136(529) | 0.2 |
| Glucose | 130(56) | 121(47) | <0.00001 |
| Hematocrit | 37.2(7.8) | 37.5(7.7) | 0.8 |
| Neutrophil | 10.2(7.4) | 8.2(5.6) | <0.00001 |
| NT-proBNP | 1463(4574) | 713(2657) | <0.0001 |
| Lactate | 1.5(1.0) | 1.4(0.9) | <0.001 |
| PaO2 | 70(41) | 68.9(39.5) | 0.1 |
| PaCO2 | 39(16) | 42(17) | 0.07 |
| PH | 7.39(0.11) | 7.39(0.1) | 0.4 |
| Procalcitonin | 0.45(0.83) | 0.24(0.48) | <0.0001 |
| Sodium | 137(5.8) | 138(5) | <0.00001 |
| WBC | 12.4(7.9) | 11(6.1) | <0.0001 |
| Clinical Characteristics | | | |
| Altered mental status | 131(10.1%) | 157(7.9%) | <0.05 |
| DBP | 68(21) | 70(22) | <0.001 |
| Pulse | 92(27) | 87(26) | <0.00001 |
| RR | 21(6) | 20(6) | <0.05 |
| Systolic BP | 125(31) | 131(32) | <0.00001 |
| Temp | 36.9(0.7) | 36.8(0.5) | <0.00001 |
| Weight | 79(32.3) | 79.8(30) | 0.2 |
| Severity Scores | | | |
| Apache III score | 65(27) | 62(28) | <0.05 |
| CURB65score | 3(1) | 3(1) | <0.00001 |
| PSI score | 115(53) | 107(50) | <0.00001 |
| 24hrs SOFA score | 4(5) | 4(4) | 0.09 |
| Steroid | 422 (32.4%) | 642(32.4%) | 0.97 |

Missing variables were excluded from the analysis.

Nonparametric Wilcoxon rank-sum test for continuous variables and the chi-square test for categorical variables.

**Table 2. Clinical outcomes based on eosinopenic status.**

| Outcomes | Eosinopenia (n = 1304) | No-eosinopenia (n = 1981) | Univariate analysis | **Multivariate analysis |
|---|---|---|---|---|
| | | | Odds ratio (95% CI), Estimate (95% CI) | |
| Primary Outcome | | | | |
| In-hospital death | 40 | 44 | (1.39, 0.90-2.2) | (1.13, 0.73-1.77) |
| 30-day mortality | 76 | 129 | (0.89, 0.66-1.19) | (0.71, 0.53-.97) |
| Secondary Outcomes | | | | |
| Need for NIV | 122 | 164 | (1.2, 0.94-1.54) | (1.04, 0.8-1.34) |
| Invasive Ventilation | 81 | 101 | (1.3, 0.96-1.75) | (0.99, 0.72-1.37) |
| Need for NIV+IMV | 73 | 57 | (2.07, 1.45-2.95) | (1.7, 1.16-2.46) |
| ICU admission | 514 | 587 | (1.55, 1.33-1.79) | (1.36, 1.16-1.59) |
| Vasopressors support | 285 | 361 | (1.26, 1.6-1.49) | (1.08, 0.90-1.29) |
| Length of hospital stay | 3.9 | 3.6 | (MD: 0.56, 0.16-0.95) | (MD: 0.30, -0.09-0.69) |

ICU; Intensive care unit, IMV; Invasive Ventilation, NIV; Non-Invasive Ventilation, MD: Mean difference.

**Multivariate analysis: adjusted for Pneumonia severity index, COPD.

ventilatory support, ICU admission, or vasopressor support), the combined pneumonia severity index and eosinopenia exhibited an AUROC of 0.725 (95% CI = 0.71–0.74), followed by pneumonia severity index (AUROC = 0.723, 95% CI = 0.71–0.74) and eosinopenia (AUROC = 0.543, 95% CI = 0.53–0.56) (p < 0.0001) (S1 and S2 Figs). Moreover no significant difference was noted when comparing the prognostic value of PSI and PSI + Eosinopenia models for 30day mortality (p: 0.12) or composite outcome (p: 0.11). The sensitivity analysis showed similar results when using eosinophil as a continuous variable.

## Discussion

In this cohort study of 3285, we evaluated the association of eosinopenia in predicting adverse outcomes in hospitalized patients with CAP. Notably, a substantial proportion (39.70%) of patients exhibited eosinopenia upon admission and had higher disease severity scores compared to the non-eosinopenic group. This highlights the potential differences in immune response, as it relates with disease severity in CAP patients, similar to the previously suggested mechanism in other infectious and inflammatory conditions [16,24–27].

Over the recent years eosinopenia has been linked to a poor prognosis and increased disease severity in various inflammatory conditions including sepsis, acute exacerbations of chronic obstructive pulmonary disease, and other infectious diseases [15–20,27]. Unfortunately the data on the prognostic value of eosinopenia specific to community acquired pneumonia is limited to few studies in patient with CAP [17,21,28].

The univariate and multivariate analysis of our cohort demonstrated a higher likelihood of ICU admission, and concurrent need of both NIV and IMV support in the eosinopenic group, emphasizing its potential as a prognostic indicator in CAP similar to the result seen in other disease states [15–20]. However, intriguingly, when subgroup analysis was performed adjusting for CRP levels in the regression model, the differences in outcomes between the groups diminished, suggesting a potential interplay between eosinopenia and other inflammatory markers in influencing disease progression and outcomes. With prior studies reporting significant association between CRP levels and clinical outcomes in CAP, this finding might reflect a direct effect modification of CRP on the relationship between eosinopenia and clinical outcomes. Further investigation is required to gain a better understanding of the underlying mechanisms that drive these associations.

In contrast to the increased risk of long term mortality associated with eosinopenia in patient hospitalized with CAP reported by Echevarria et al and Mao et al [17,21], our study revealed a decreased adjusted odds of 30-day mortality in patients with eosinopenia after adjusting for PSI and COPD. However, no substantial difference in mortality was observed for in-hospital or 30-day mortality in the subgroup analysis of patients < 65 years old and those with CRP after adjusting for PSI, COPD and CRP level.

Although an increase in 30-day mortality was observed after adjusting for steroid use in the eosinopenic group, a decrease in mortality was noted after adjustment for early steroid use. This suggests a potential benefit of early steroid use (within 24 hours) in severe CAP similar to prior studies [29,30], as compared to late steroid use If this finding is replicated in other studies, eosinopenia could be useful in r identifying patients who may benefit from early steroid use. Furthermore, the sensitivity analyses exploring various eosinophil levels consistently mirrored the initial findings, and the prognostic value of the PSI remained superior to eosinopenia with a nonsignificant increase when both are combined.

In comparison to the previously published studies the strengths of our study included the large sample size (n = 3285) offering a robust sample size for analysis. The exclusion of patients that received steroids before the eosinophil level blood draw addresses the potential effect of steroids on eosinophil count. The use of ICD codes with further verification of a diagnosis of community acquired pneumonia by clinical note search helped with decreasing the risk of misclassification [23].

As is common in retrospective observational studies, one of our study's limitations was the utilization of electronic health records to obtain data which may have introduced potential biases due to incomplete or missing variables and the inability to account for unmeasured confounders. The study design could not assess for the influence of unmeasured confounders, such as specific microbial etiologies, individual variations in immune responses, vaccination history, concomitant use of antihistamines agents, prior steroid use before admission or medication effects although the larger sample size of our cohort would have help reduce the impact of this limitation. Furthermore, the absence of mechanistic investigations also restricts our understanding of the underlying pathophysiological mechanisms linking eosinopenia to CAP outcomes.

Given the significant impact of community acquired pneumonia, further efforts are needed to gain a better understanding and insights into the factors influencing disease prognosis. Our study will add to the limited existing body of literature on the association of eosinopenia in CAP, challenging previous assumptions and emphasizing the need for a comprehensive understanding of the multifactorial nature of prognosis in patients hospitalized with CAP infection.

## Conclusion

Contrary to previously published data, our analysis did not demonstrate an association between eosinopenia and increased mortality risk in hospitalized patients with CAP, highlighting the complexity of CAP prognosis. Further investigations and comprehensive approaches are warranted to elucidate the intricate interplay of variables contributing to the prognosis of hospitalized CAP patients.

## Supporting information

**S1 Fig. Prognostic accuracy of eosinopenia and pneumonia severity index in predicting 30-day mortality.**
(DOCX)

**S2 Fig. Prognostic accuracy of eosinopenia and pneumonia severity index in predicting composite secondary outcomes.**
(DOCX)

**S1 File. Dataset.**
(XLSX)

**S1 Table. Clinical outcomes based on eosinopenic status in subgroup of patients with CRP.**
(DOCX)

**S2 Table. Clinical outcomes based on eosinopenic status in subgroup of patients less than 65 years old.**
(DOCX)

**S3 Table. Mortality outcomes based on eosinopenic status after adjustment for steroid treatment during hospitalization.**
(DOCX)

**S4 Table. Mortality outcomes based on eosinopenic status a after adjustment for early (within 24 hours) steroid treatment.**
(DOCX)

## Acknowledgments

We would like to acknowledge the Anesthesia Clinical Research Unit study coordinator, Alberto Marquez for his help with data extraction.

## Author contributions

**Conceptualization:** Wigdan Farah, Yewande E. Odeyemi.

**Data curation:** Wigdan Farah, Zhen Wang, Yewande E. Odeyemi.

**Formal analysis:** Wigdan Farah, Zhen Wang.

**Funding acquisition:** Yewande E. Odeyemi.

**Methodology:** Wigdan Farah, Zhen Wang, Ognjen Gajic, Yewande E. Odeyemi.

**Software:** Wigdan Farah, Zhen Wang.

**Supervision:** Ognjen Gajic, Yewande E. Odeyemi.

**Validation:** Wigdan Farah, Zhen Wang, Yewande E. Odeyemi.

**Visualization:** Ognjen Gajic, Yewande E. Odeyemi.

**Writing – original draft:** Wigdan Farah, Zhen Wang, Ognjen Gajic, Yewande E. Odeyemi.

**Writing – review & editing:** Wigdan Farah, Zhen Wang, Ognjen Gajic, Yewande E. Odeyemi.

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
