## [Decision Letter · Decision Letter 0]

14 Jun 2024

PONE-D-24-12781Eosinopenia as a Predictor of Clinical Outcomes in Hospitalized Patients with Community-Acquired Pneumonia: A Retrospective Cohort StudyPLOS ONE

Dear Dr. Farah,

Thank you for submitting your manuscript to PLOS ONE. After careful consideration, we feel that it has merit but does not fully meet PLOS ONE’s publication criteria as it currently stands. Therefore, we invite you to submit a revised version of the manuscript that addresses the points raised during the review process.

We look forward to receiving your revised manuscript.

Kind regards,

Nosheen Nasir

Academic Editor

PLOS ONE

Journal Requirements:

this research was supoorted by grant number K23HL168212 (National Institute of Health, recip-ient: YO), Robert D. and Patricia E. Kern Center for the Science of Health Care Delivery, Mayo Clinic (recipient: YO) and Robert A. Winn Diversity in Clinical Trials Career Development Award (recipient: YO).

Reviewers' comments:

Reviewer's Responses to Questions

**Comments to the Author**

1. Is the manuscript technically sound, and do the data support the conclusions?

Reviewer #1: Yes

Reviewer #2: Partly

2. Has the statistical analysis been performed appropriately and rigorously? 

Reviewer #1: Yes

Reviewer #2: Yes

3. Have the authors made all data underlying the findings in their manuscript fully available?

Reviewer #1: Yes

Reviewer #2: Yes

4. Is the manuscript presented in an intelligible fashion and written in standard English?

Reviewer #1: Yes

Reviewer #2: No

5. Review Comments to the Author

**Reviewer #1:**  The study is interesting considering the fact that eosinopenia may be helpful in identifying patients with high risk of mortality and optimising management strategies accordingly. I have certain queries for the authors:

1) What exactly were the criteria for labelling as CAP? Please specify precisely the clinical, laboratory and radiology criteria for defining CAP in your study?

Was an X ray or CT scan used as the radiological test? What radiological evidence was used to call it CAP?Lobar / segmental consolidation/ interstitial opacities/ ground glass opacities?

2) Were patients who were on antihistamines excluded from the study as that could have interfered with the eosinophil count?

3) What proportion of the patients were referred from the EMD and how many were admitted from the OPD?

4) Was there a correlation between the two groups and the D dimer , pro calcitonin , pro BNP, baseline PaCO2 levels?

5) What were the first line antibiotics used in the patients and was there any specific antibiotic policy adopted for the patients?

6) Was there a seasonal variation in the pattern of CAP observed in the patients?

7) Was history of vaccination like pneumococcal or influenza taken into account as these could have a protective role in CAP?

8) Line no. 137 seems to be incomplete.

Regards,

Dr Supriya.

**Reviewer #2:**  Respected author,

The manuscript written , titled as “Eosinopenia as a predictor of clinical outcomes in hospitalized patients with community-acquired pneumonia: a retrospective cohort study” addresses an important issue, addresses an important issue, inflammatory marker as predictor of outcomes in community acquired pneumonia. It highlights that the eosinopenia is a strong predictor of poor outcomes for patients with CAP however varies factors such as COPD and history of steroid intake may influence it.

There were a few concerns raised by the reviewer which may need to be addressed by the authors for better understanding and scientific soundness of the article. As researchers have honestly highlighted, several bias like selection bias, recall bias, missing data , and the unmeasured underlying disease status are a major limitation of the retrospective electronic data collection.

Errors have been noted in the representation of statistical data at various instances. Authors furthermore requested to rectify the various spelling and grammatical errors that have been overlooked. Authors are urged to revise the numbering of the references as per the journals recommendation and as per standard practice.

6. PLOS authors have the option to publish the peer review history of their article (what does this mean? ). If published, this will include your full peer review and any attached files.

**Do you want your identity to be public for this peer review?** For information about this choice, including consent withdrawal, please see our Privacy Policy .

Reviewer #1: No

Reviewer #2: No

---

## [Author Response · Author response to Decision Letter 1]

22 Jul 2024

Thank you for the comments. Our response to the reviewer and editors comments are detailed in our response to the reviewers file.

Many Thanks

Wigdan

---

## [Decision Letter · Decision Letter 1]

17 Sep 2024

PONE-D-24-12781R1Eosinopenia as a Predictor of Clinical Outcomes in Hospitalized Patients with Community-Acquired Pneumonia: A Retrospective Cohort StudyPLOS ONE

Dear Dr. Farah,

Thank you for submitting your manuscript to PLOS ONE. After careful consideration, we feel that it has merit but does not fully meet PLOS ONE’s publication criteria as it currently stands. Therefore, we invite you to submit a revised version of the manuscript that addresses the points raised during the review process.

We look forward to receiving your revised manuscript.

Kind regards,

Nosheen Nasir

Academic Editor

PLOS ONE

Journal Requirements:

Additional Editor Comments:

**Please address comments by the reviewers specified in email.**

Reviewers' comments:

Reviewer's Responses to Questions

**Comments to the Author**

1. If the authors have adequately addressed your comments raised in a previous round of review and you feel that this manuscript is now acceptable for publication, you may indicate that here to bypass the “Comments to the Author” section, enter your conflict of interest statement in the “Confidential to Editor” section, and submit your "Accept" recommendation.

Reviewer #1: (No Response)

Reviewer #3: (No Response)

Reviewer #4: (No Response)

2. Is the manuscript technically sound, and do the data support the conclusions?

Reviewer #1: Yes

Reviewer #3: Partly

Reviewer #4: Partly

3. Has the statistical analysis been performed appropriately and rigorously? 

Reviewer #1: Yes

Reviewer #3: I Don't Know

Reviewer #4: Yes

4. Have the authors made all data underlying the findings in their manuscript fully available?

Reviewer #1: Yes

Reviewer #3: Yes

Reviewer #4: Yes

5. Is the manuscript presented in an intelligible fashion and written in standard English?

Reviewer #1: Yes

Reviewer #3: Yes

Reviewer #4: Yes

6. Review Comments to the Author

Reviewer #1: All comments except one have been addressed. There was no x ray correlation in this study which I thought was quite important to ascertain the diagnosis and to assess the severity of pneumonia. It would have been nice to know if the eosinopenia was related to the x ray pattern that is unilobar , multilobar pneumonia. Also, it would have been good to know how many viral and bacterial pneumonias were there in the study. I understand it is a retrospective study and the other comments have been included in the discussion section as limitations.

Reviewer #3: The manuscript is improved, but I've still some concern about the main message and the influence of steroid treatment.

It's important to clear that eosinopenia hasn't yet been demonstrated to be an independent risk factor for outcome in community-acquired pneumonia or severe infections. The initial sentence in the Abstract is not true and it should be erased. Also the conclusion of the Abstract should be tuned down in accordance with the results. Please, cite and discuss Lin Y, et al. Silent existence of eosinopenia in sepsis: a systematic review and meta-analysis. BMC Infect Dis. 2021; 21: 471. Actually, eosinopenia was frequently associated to severe infection and sepsis, but it has no shown superiority in comparison with conventional biomarkers. Also this retrospective study is in accordance with it. Another point is the influence of eosinopenia on steroid treatment results. It is well known that several studies showed a significant positive effect of corticosteroid therapy on mortaly from severe CAP (Confalonieri M, et al. Am J Respir Crit Care Med 2005 and Dequin PF, et al. N Engl J Med 2023). Please, cite both papers and statistically analyze in this retrospective study population if the presence of eosinopenia influences the response to corticosterois on mortality or other outcomes.

Reviewer #4: Dear Authors,

Although I believe this article does address important information and adds to the body of knowledge surrounding the association between eosinopenia and clinical outcomes in hospitalized patients with CAP. I had many outlying questions that were not answered, such as:

1. This study attempts to analyze Eosinopenia is an independent predictor of adverse outcomes and a marker of severity in bacterial infections, describing the association between eosinopenia and clinical outcomes in hospitalized patients with CAP and looking for associations with increased risk of mortality in hospitalized patients with community-acquired pneumonia. The authors used study design retrospective analysis of adults hospitalized with CAP from January 2009 to December 2019 at a large academic medical center, with Participants 3,285 patients, about 1,304 (39.70%) were classified as eosinopenia. Data collected from demographics, disease severity, comorbidities, smoking history, inflammatory markers, blood eosinophil levels, mortality, length of hospital stay, ICU admission, and need for mechanical ventilation. Statistical methods used nonparametric Wilcoxon rank sum test for continuous variables, chi-square test for categorical variables, and logistic regression with robust standard errors for outcome associations.

Several previous studies…A study by Mao et al [17] showed an association between eosinopenia and increased risk of long-term mortality in hospitalized patients with CAP but with a concurrent diagnosis of acute exacerbation of COPD. Notably, there was a significant difference in total steroid exposure between the eosinopenic and non-eosinopenic groups in this study. Similar results were also reported in a second study by Echevarria et al [21] which focused on a group of hospitalized patients with a diagnosis of CAP, excluding patients with an underlying diagnosis of chronic respiratory disease in an attempt to address the effects of steroids on eosinophils. Although both studies reported a significant increase in CRP in the eosinopenic group compared with the non-eosinopenic group, indicating a higher disease severity in the eosinopenic group, CRP levels were not considered in the analysis. In addition, both studies were limited by sample size. Conclusion in this study, eosinopenia was not associated with an increased risk of death in hospitalized patients with community-acquired pneumonia. This is contrary to previously published data and highlights the complexity of CAP prognosis. Can you explain it ?

2. The authors evaluated the association of eosinopenia in predicting adverse outcomes in 3285 hospitalized patients with CAP. Most (39.70%) patients showed eosinopenia on admission and had higher disease severity scores compared to the non-eosinopenic group. Inflammatory Markers and Disease Severity with Eosinopenic patients having higher C-reactive protein (CRP) levels and higher disease severity scores compared to non-eosinopenic patients. After adjusting for disease severity, chronic obstructive pulmonary disease (COPD), and CRP, the authors found no significant differences in In-Hospital Mortality, ICU Admission, and Mechanical Ventilation. What is the reason?

3. The manuscript provides sufficient context for the tables. Table titles appear to describe the content and context. Tables are consistent throughout the manuscript. Consistent color schemes, fonts, and formatting make the manuscript more professional and easy to follow. Tables directly support the conclusions of the manuscript. No Unnecessary Information.

4. The method chosen is the most appropriate to test the hypothesis or answer the research question posed. In this study, the sample size and amount of data collected are sufficient to support the conclusions drawn. The author does not mention limitations and potential sources of error in the study, avoiding bias and shortcomings or any factors that may affect the results? So that the methods described are clear and comprehensive enough for other researchers to repeat the study and verify the results.

5. The data presented in the results section directly support the conclusions. A clear relationship between the results and conclusions ensures that the author's interpretation is justified by the data. The conclusions drawn are appropriate to the results and scope of the study.

6. In conclusion, the authors found that patients with eosinopenia on admission were generally sicker, with more severe disease markers and higher levels of inflammation and stress markers. Despite these differences in clinical characteristics, the primary outcomes of in-hospital mortality and 30-day mortality were not significantly different between the eosinopenia and non-eosinopenia groups. However, eosinopenia patients were more likely to require non-invasive and invasive mechanical ventilation and had higher odds of requiring ICU admission and vasopressors, reflecting the need for more severe disease management. Length of hospital stay was longer for eosinopenia patients, although these findings were less consistent after multivariate adjustment.

7. PLOS authors have the option to publish the peer review history of their article (what does this mean? ). If published, this will include your full peer review and any attached files.

**Do you want your identity to be public for this peer review?** For information about this choice, including consent withdrawal, please see our Privacy Policy .

Reviewer #1: No

Reviewer #3: No

Reviewer #4: No

---

## [Author Response · Author response to Decision Letter 2]

30 Oct 2024

Please find our detailed responses to the reviewer and editor comments in our response to the reviewer file

---

## [Decision Letter · Decision Letter 2]

11 Nov 2024

Eosinopenia as a Predictor of Clinical Outcomes in Hospitalized Patients with Community-Acquired Pneumonia: A Retrospective Cohort Study

PONE-D-24-12781R2

Dear Dr. Farah,

We’re pleased to inform you that your manuscript has been judged scientifically suitable for publication and will be formally accepted for publication once it meets all outstanding technical requirements.

Kind regards,

Chinh Quoc Luong, MD., PhD.

Academic Editor

PLOS ONE

Additional Editor Comments (optional):

All comments have been addressed.

Reviewers' comments:

Reviewer's Responses to Questions

**Comments to the Author**

1. If the authors have adequately addressed your comments raised in a previous round of review and you feel that this manuscript is now acceptable for publication, you may indicate that here to bypass the “Comments to the Author” section, enter your conflict of interest statement in the “Confidential to Editor” section, and submit your "Accept" recommendation.

Reviewer #1: All comments have been addressed

Reviewer #4: All comments have been addressed

2. Is the manuscript technically sound, and do the data support the conclusions?

Reviewer #1: Yes

Reviewer #4: Yes

3. Has the statistical analysis been performed appropriately and rigorously? 

Reviewer #1: Yes

Reviewer #4: Yes

4. Have the authors made all data underlying the findings in their manuscript fully available?

Reviewer #1: Yes

Reviewer #4: Yes

5. Is the manuscript presented in an intelligible fashion and written in standard English?

Reviewer #1: Yes

Reviewer #4: Yes

6. Review Comments to the Author

Reviewer #1: Most of the comments have been addressed and have been discussed as limitations of the study.can be accepted if other comments have also been addressed to satisfaction.

Reviewer #4: The manuscript presents a valuable contribution. The authors effectively [summarize strengths, such as methodology, data analysis, or findings. I am happy that the revised manuscript has thoroughly addressed the issues raised in the original manuscript.

7. PLOS authors have the option to publish the peer review history of their article (what does this mean? ). If published, this will include your full peer review and any attached files.

**Do you want your identity to be public for this peer review?** For information about this choice, including consent withdrawal, please see our Privacy Policy .

Reviewer #1: No

Reviewer #4: No

---

## [Editor Report · Acceptance letter]

PONE-D-24-12781R2

PLOS ONE

Dear Dr. Farah,

I'm pleased to inform you that your manuscript has been deemed suitable for publication in PLOS ONE. Congratulations! Your manuscript is now being handed over to our production team.

Kind regards,

on behalf of

Assoc. Prof. Chinh Quoc Luong

Academic Editor

PLOS ONE